# Anti-Inflammatory, Anti-Diabetic, Anti-Oxidant and Cytotoxicity Assays of South African Herbal Teas and Bush Tea Blends

**DOI:** 10.3390/foods11152233

**Published:** 2022-07-27

**Authors:** Florence Malongane, Lyndy Joy McGaw, Oyinlola Oluwunmi Olaokun, Fhatuwani Nixwell Mudau

**Affiliations:** 1Department of Life and Consumer Sciences, College of Agriculture and Environmental Sciences, University of South Africa, Private Bag X6, Florida 1710, South Africa; 2Phytomedicine Programme, Department of Paraclinical Sciences, University of Pretoria, Private Bag X04, Onderstepoort, Pretoria 0110, South Africa; lyndy.mcgaw@up.ac.za; 3Department of Biology, Sefako Makgatho Health Sciences University, Molotlegi Street, Ga-Rankuwa, Pretoria 0204, South Africa; oyinolaokun@yahoo.com; 4School of Agriculture, Earth and Environmental Sciences, University of Kwazulu Natal, Private Bag X01, Scottsville, Pietermaritzburg 3209, South Africa; mudauf@ukzn.ac.za

**Keywords:** anti-oxidant, anti-inflammatory, anti-diabetic, cytotoxicity

## Abstract

South Africa is home to a variety of herbal teas, such as bush tea (*Athrixia phylicoides* DC.), honeybush tea (*Cyclopia intermedia* E. Mey and *C. subternata* Vogel), special tea (*Monsonia burkeana* Planch. ex Harv.), and rooibos tea (*Aspalathus linearis* (Burm.f.) R. Dahlgren) that are known to possess anti-oxidant, anti-inflammatory and anti-diabetic properties. The objective of this study was to determine the in vitro anti-oxidant activity of selected tea blends using 2,2′-azino-bis(3-ethylbenzthiazoline-6-sulfonic acid) (ABTS) and 1,1-diphenyl-2-picrylhydrazyl (DPPH) assays, as well as to assess their anti-inflammatory properties using the 15-lipoxygenase inhibitory assay. Furthermore, the study measured glucose utilisation in C2C12 myotubes. Lastly, 3-(4, 5-dimethylthiazol-2-yl)-2, 5-diphenyltetrazolium bromide (MTT) assay was used to test the safety of the tea extracts on Vero cells (African green monkey kidney cell line). Special tea and its blend with bush tea exhibited potent anti-oxidant and anti-inflammatory activity. The blending of bush tea with special tea at different ratios resulted in increased anti-oxidant activity. Although special tea had a level of cell toxicity, its toxicity was lowered during blending. All of the tea samples showed anti-diabetic effects, although with less potency as compared to insulin. The current investigation supports the use of blended herbal teas, and the positive anti-inflammatory effect of special tea warrants further research.

## 1. Introduction

Bush tea (*Athrixia phylicoides* DC.), special tea (*Monsonia burkeana* Planch. ex Harv.), rooibos tea (*Aspalathus linearis* (Burm.f.) R. Dahlgren) and honeybush tea (*Cyclopia intermedia* E. Mey and *C. subternata* Vogel) are South African herbal teas all of which are known to possess anti-oxidant [1], anti-inflammatory [2] and anti-diabetic properties [3,4,5].

South African indigenous people have used Bush tea as a treatment for boils, coughs, colds, headaches and also as a remedy for aphonia [6]. The ground and pulverised bush tea leaves have also been used as an ointment for treating skin eruptions and cleansing infected wounds [7]. Special tea is reported to treat sexually transmitted diseases, used for blood cleansing and improve erectile dysfunction [8]. Rooibos and honeybush are among the teas available in international markets [9,10]. Rooibos tea has been used to manage stomach cramps, cure insomnia, and as a treatment for sunburn, rashes, and eczema [11]. On the other hand, honeybush tea is known to treat constipation, calm the central nervous system, and be an expectorant in pulmonary diseases [12].

Herbal teas contain polyphenols responsible for their anti-oxidant and anti-inflammatory activities [13]. For instance, rooibos tea contains aspalathin, luteolin, nothofagin, orientin, vitexin, isoorientin, isovitexin, chrysoeriol, rutin, isoquercitrin, quercetin, vanillic acid, syringic acid, cinnamic acids, p-coumaric acid, ferulic acid and caffeic acid. Aspalathin, the main compound in rooibos tea, possesses anti-oxidant activities [14,15].

Anti-oxidant compounds are said to protect the cell membrane against harm caused by reactive oxygen species (ROS). In diabetic conditions, anti-oxidants inhibit the glycation process, a reaction known to form advanced glycation end-products (AGEs) [16]. The accumulation of AGEs leads to macro-and micro-vascular complications in diabetic patients [17]. In addition, herbal teas have been shown to have hypoglycaemic effects by enhancing glucose uptake by adipose and muscle tissues and inhibiting glucose production, thus assisting in the management of diabetes mellitus [18].

Diabetes mellitus is a non-communicable disease known as one of the top ten leading causes of disability globally. It is estimated to affect about 366 million people in the world [19]. Chellan [20] reported bush tea’s hypoglycaemic effect and attributed its effectiveness to its phenolic compound composition. The anti-diabetic effects of rooibos tea extract or its component aspalathin have been confirmed both in vitro and in vivo [5,21]. Rooibos water extract decreased the formation of AGEs after incubation of glucose and serum albumin, and the effect was attributed to the presence of compounds with anti-oxidant properties [22]. Aspalathus increased glucose uptake by L6 myotubes in culture and improved impaired glucose tolerance in type 2 diabetic model db/db mice [5]. Similarly, unfermented honeybush tea was found to offer a protective effect on pancreatic β-cells of streptozotocin (STZ)-induced diabetic Wistar rats by improving glucose tolerance [23]. Mangiferin, a major compound in honeybush tea, lowered blood glucose in KK-Ay mice [24]. However, no data on the anti-diabetic effects of special tea are currently available.

The inhibition of enzymes involved in inflammation is one of the pathways of anti-inflammatory activity. Polyphenols in plant extracts are thought to be able to inhibit the biosynthesis of prostaglandins, end-products in the cyclooxygenase (COX) and lipoxygenase (LOX) pathways of immune responses [25]. During inflammation, arachidonic acid is metabolised through the COX or LOX pathway [25]. Baba [2] reported the anti-inflammatory properties of rooibos tea in a DSS-induced colitis rat model supplemented with rooibos tea which had increased serum superoxide dismutase compared to the control group. There was no anti-inflammatory activity shown by the methanol extract of bush tea; the activity was only observed in bush tea’s essential oil [26]. A consideration for blending bush tea with other teas is that this could increase its anti-inflammatory activity by offering a synergistic effect.

Although ample research has been conducted on the anti-oxidant, anti-diabetic, and anti-inflammatory of rooibos and honeybush tea, together with their components, there are still insufficient studies on bush tea and the anti-diabetic and anti-inflammatory activities of special tea. Furthermore, no study has investigated the potential synergistic effect of blending bush with honeybush, rooibos and special tea for health promotion. Thus, this study’s objective was to determine the anti-oxidant, anti-inflammatory and anti-diabetic activities and the cytotoxicity of hot water extracts of the bush, honeybush, special and rooibos tea. Additionally, the researchers investigated bush tea’s potential synergistic effect when blended with the honeybush, special and rooibos tea at different ratios.

## 2. Materials and Methods

### 2.1. Plant Material

Tea samples selected for the study included 100% bush (B100), 100% fermented honeybush from species *Cyclopia subternata* (H100), 100% special (S100) and 100% fermented rooibos tea (R100) as well as nine blends of bush tea, namely 50% bush tea plus 50% honeybush tea (BH50:50), 50% bush tea plus 50% special tea (BS50:50), 50% bush tea plus 50% rooibos tea (BR50:50), 75% bush tea plus 25% honeybush tea (BH75:25), 75% bush tea plus 25% special tea (BS75:75%), 75% bush tea plus 25% rooibos tea (BR75:25), 25% bush tea plus 75% bush tea (BH25:75), 25% bush tea plus 75% special tea (BS25:75) and 25% bush tea plus 75% rooibos tea (BR25:75). Rooibos and honeybush tea were purchased from local markets in Gauteng Province, South Africa. Bush tea samples were collected from the Agricultural Research Council (ARC) in Pretoria, South Africa, while special tea leaves were collected in Hartbeespoort, Northwest Province, South Africa. The four South African herbal teas were selected based on their local availability.

### 2.2. Extraction

Water was used as an extraction solvent for all tea samples. In brief, 4 g of each tea sample was placed in 50 mL centrifuge tubes, followed by adding 40 mL of boiled (100 °C) deionised water as described by Eloff [27]. The mixtures were mechanically shaken overnight, after which they were sonicated for 10 min and then centrifuged for 10 min. The resultant liquid was filtered into a beaker through Whatman No. 1 filter paper with an 11 µm pore size. The filtered tea extracts were poured into centrifuge tubes, stored overnight at −80 °C, and then freeze-dried using a Labconco Freezone 4.5 Benchtop Freeze Dryer (Labconco corporation, Kansas City, MO, USA). The resultant extract was then used for all subsequent assays.

### 2.3. Assays

#### 2.3.1. Cytotoxicity Assay

The cytotoxicity of the aqueous tea extracts against Vero cells (African green monkey kidney cell line) purchased from the American Type Culture Collection (ATCC CCL-81) (Manassas, VA, USA) was determined using the 3-(4,5-dimethylthiazol-2-yl)-2, 5-diphenyltetrazolium bromide (MTT) reduction assay as described by Mosmann [28]. The dried tea extracts were re-suspended in dimethyl sulfoxide (DMSO) to a 100 mg/mL concentration. Solutions of varying concentrations (1 000, 500, 250, 125 μg/mL) were prepared in cell culture growth medium by serially diluting the stock solution.

Cells of a sub-confluent culture were harvested and centrifuged at 200× *g* for 5 min, and re-suspended in minimal essential growth medium (MEM, Whitehead Scientific, Johannesburg, South Africa) to 5 × 10^4^ cells/mL. The medium was supplemented with 0.1% gentamicin (Virbac, Centurion, South Africa) and 5% foetal calf serum (Highveld Biological, Sandton, Johannesburg, South Africa). A total of 200 µL of the cell suspension was pipetted into each well of columns 2 to 11 of a sterile 96-well microtitre plate. Thereafter, 200 mL of MEM was added to the wells of columns 1 and 12 to maintain humidity and minimise the “edge effect”. The 96-well plates were incubated for 24 h at 37 °C in a 5% CO_2_ incubator until the cells were in the exponential phase of growth. The MEM was removed from the cells and replaced with 200 µL of test compound at different concentrations, in quadruplicate. Both serial dilutions of the test extracts and compounds were prepared in MEM. The cells were disturbed as little as possible during the aspiration of medium and addition of the test substance. The 96-well microtitre plates were incubated at 37 °C in a 5% CO_2_ incubator for 48 h with a test compound or extract. Doxorubicin chloride (Pfizer Laboratories, Johannesburg, South Africa) was used as a positive control.

After incubation, the MEM with test substance was removed from the cells, which were then washed with 150 μL of phosphate-buffered saline (PBS, Whitehead Scientific, Johannesburg, South Africa) and replaced with 200 µL of fresh MEM. Subsequent to this, 30 µL of MTT (Sigma-Aldrich, Johannesburg, South Africa, stock solution of 5 mg/mL in PBS) was added to each well, and the plates were incubated for a further 4 h at 37 °C. Following incubation with MTT, the medium in each well was removed without disturbing the MTT crystals in the wells. The MTT formazan crystals were dissolved by adding 50 µL of DMSO to each well. The plates were gently shaken until the MTT solution was dissolved. The MTT reduction was determined immediately by measuring absorbance in a Biotek Synergy Analytical and Diagnostic Products, Johannesburg, South Africa) at a wavelength of 570 nm and a reference wavelength of 630 nm. The wells in column 1 containing MEM and MTT, but no cells, and were used to blank the plate reader. The median lethal concentration (LC_50_) was calculated and defined as the concentration of the test compound, resulting in a 50% reduction in absorbance compared to untreated cells.

#### 2.3.2. Anti-Oxidant Assays

##### The 2,2-Azino-Bis (3-Ethylbenzothiazoline-6-sulfonic Acid) Diammonium Salt (Abts) Anti-Oxidant Assay

The tea extracts’ quantitative ABTS radical scavenging capacity was measured using the 96-well microtitre plate method described by Re et al. [29]. Briefly, all of the tea sample extracts were reconstituted in methanol at a concentration of 1 mg/mL, whereafter 40 µL of methanol were added to all the wells of the 96-well plate. Thereafter, 40 µL of tea samples or control were added in row A. The mixture was serially diluted until row H and the remaining 40 µL discarded. Each tea sample or control was added to four columns. One hundred and sixty microlitres (160 µL) of a 25 µL/mL of ABTS solution was added to the first two columns, after which 160 µL of methanol was added to the next two columns. The plates were incubated in the dark for 5 min. Ascorbic acid and trolox were used as positive controls, while methanol was used as the negative control and tea extracts without ABTS as blanks. The absorbance was read at 734 nm. The percentage of ABTS inhibition was calculated using the formula:(1)$$\%\;\text{scavenging}\;\text{activity}=\left[\frac{\text{absorbance}\;\text{of}\;\text{control}-\text{absorbance}\;\text{of}\;\text{the}\;\text{sample}}{\text{absorbance}\;\text{of}\;\text{control}}\right]\times 100$$

The results were presented as IC_50_.

##### The 2,2-Diphenyl-1-picrylhydrazyl (DPPH) Anti-Oxidant Scavenging Assay

The DPPH radical scavenging activity of methanol extracts was determined using the method explained by Brand-Williams et al. [30]. The method measures the ability of an anti-oxidant compound to scavenge reactive oxygen species (ROS) by either donating hydrogen or transferring an electron [31]. All of the extracts were reconstituted in methanol to a starting concentration of 1 mg/mL. Methanol (40 μL) was added to all the 96 wells of a microtitre plate, while 40 μL of the sample or control were pipetted into row A and thereafter serially diluted up to row H of the 96-well microplate.

One hundred and sixty microlitres (160 μL) of the methanolic solution of DPPH (25 µg/mL) were added to the first two columns, and the same volume of methanol was added to the subsequent two columns for each sample and control. The mixture was then incubated for 30 min, and the final absorbance was read at 517 nm using a Biotek microplate reader (Synergy 2 Multi-Mode Reader, BioTek Instruments, Winooski, VT, USA). The experiment was performed in duplicate and repeated twice. Ascorbic acid and trolox served as positive controls at a 1 mg/mL concentration. The results were expressed as the total anti-oxidant concentration necessary to reduce the initial DPPH absorbance by 50%.

#### 2.3.3. Anti-Inflammatory Assays

A spectrophotometric assay for the determination of LOX activity was as described by Pinto [32], with slight modifications. The assay is based on measuring the formation of the complex Fe^3+^/xylenol orange, which is read using spectrophotometer at 560 nm. The substrate linoleic acid (final concentration 140 μM) was prepared in Tris-HCl buffer (50 mM, pH 7.4). All crude tea extracts (10 mg/mL) were prepared in 100% DMSO and again diluted to 2 mg/mL in Tris-HCl buffer. Forty microlitres of the enzyme 15-LOX, diluted in ice-cold Tris-HCl buffer (final concentration 0.2 U/mL), were mixed with twenty µL of different concentrations (100 to 0.78 μg/mL) of test samples or positive control (quercetin) at 25 °C for 5 min, whereupon forty µL of linoleic acid was added to the mixture, which was then again incubated in the dark at 25 °C for 20 min. The assay was paused by adding 100 μL of ferrous oxidation−xylenol orange (FOX) reagent consisting of 30 mM sulphuric acid, xylenol orange (100 mM), and iron(II) sulphate (100 mM) in methanol/water (9:1). Only 15-lipoxygenase (15-LOX) solution and buffer were pipetted into the wells for the negative control. Blanks contained the enzyme LOX during incubation and the substrate (linoleic acid) was added to the FOX reagent. The lipoxygenase inhibitory activity was determined by calculating the percentage of the inhibition of hydroperoxide production from the changes in absorbance values at 560 nm after 30 min at 25 °C as shown in the formula below:(2)$$15-\text{LOX}\;\text{inhibition}\;\left(\%\right)=\left[\frac{\text{absorbance}\;\text{of}\;\text{control}-\text{absorbance}\;\text{of}\;\text{the}\;\text{sample}}{\text{absorbance}\;\text{of}\;\text{control}}\right]\times 100$$

#### 2.3.4. Anti-Diabetic Assays

One hundred milligrams (100 mg) of each tea extract were dissolved in 1 mL of dimethyl sulfoxide (DMSO) for the assay. The glucose utilisation assay of the selected herbal teas was performed following the method described by Olaokun [33] and Yin [34]. Briefly, previously cultured C2C12 cells in RPMI-1640 medium (Roswell Park Memorial Institute, Gibco, NY, USA) supplemented with 10% foetal bovine serum (FBS) and 1% streptomycin/penicillin solution at about 70% confluence were seeded (25,000 cells/mL) into the wells of a 96-well microtitre plate. The 2-day post-confluent cell medium was replaced by the differentiating medium (RPMI-1640 supplemented with 2% FBS) and incubated at 37 °C. After 6 days, the medium was further removed and replaced with 100 μL of RPMI-1640 supplemented with 0.25% BSA containing plant extracts at different concentrations (15.63, 31.25, 62.5, 125, 250 and 500 µg/mL) after which the plates were further incubated for 1 h. After the 1 h incubation period, the glucose concentration in the medium was determined by the glucose oxidase method (Sigma GAGO 20 test kit, St Louis, MO, USA) with absorbance measured at 540 nm. Insulin (0.01, 0.1 and 1 μM) served as the positive control and 0.5% DMSO as solvent control. All of the assays were conducted in triplicate and repeated on two occasions.

### 2.4. Data Presentation and Statistical Analysis

Data were expressed as mean ± standard error of the mean (SEM). The 50% inhibitory concentrations (IC_50_) were obtained by using the non-linear regression curve of the percentage (15-LOX) inhibition against the logarithm of concentrations tested. Cell viability assays were represented as mean ± S.E. Each test was conducted in triplicate. One-way analysis of variance (ANOVA) undertaken in SPSS 20 (IBM Corp., Armonk, NY, USA) was used for the statistical analysis of C2C12 muscle cell glucose utilisation activity induced by extracts and considered significantly different at *p* < 0.05. When significance was found, the location of significance was determined by Tukey’s HSD (honest significant difference) multiple comparison post hoc tests.

## 3. Results and Discussion

### 3.1. Cytotoxicity Assays

The cytotoxicity of the tea extracts is presented in Figure 1. The LC_50_ values were measured at concentrations of 1000 µg/mL, 500 µg/mL, 250 µg/mL and 125 µg/mL. The extracts of special tea were found to have the highest toxicity (LC_50_ < 1000 µg/mL). According to Clarkson’s toxicity criterion, Bush tea extract (LC_50_ > 1000 µg/mL) was considered non-toxic, as extracts with LC_50_ values above 1 000 μg/mL are classified as non-toxic [35]. The results concur with the findings of McGaw et al. [36], who reported that the aqueous solution of bush tea had negligent toxicity, with LC_50_ greater than 1000 µg/mL. To the best of our knowledge, the present study is the first to evaluate special tea’s cytotoxicity.

Apart from bush tea, all extracts were found to have LC_50_ values at a concentration greater than 500 μg/mL. However, at lower concentrations (125 μg/mL), all tea extracts were considered non-toxic except for special tea and BS75:25 with 50.07% and 39.37% cell viability. Honeybush and bush tea extracts are considered to have medium toxicity with LC_50_ at a concentration of 250 µg/mL [35]. Blending bush tea and special tea at 50:50 and 25:75 ratios at a concentration of 125 µg/mL resulted in reduced cytotoxicity with 66.66% and 76.80% of viable cells, respectively. The result confirms the hypothesis by Kiyohara [37] that combining herbs has the potential to reduce the adverse effects of the individual herbs, thus increasing their safety of use. Blending bush tea with honeybush, special, and rooibos tea compromised the safety of bush tea, indicating that a toxic plant species can compromise the safety or non-toxicity of plants.

### 3.2. Anti-Oxidant Activities

The combination of DPPH and ABTS assays allows for the exploration of two different pathways; the electron transfer-based assay, which measures the ability to transfer an electron and the hydrogen atom transfer assay, which involves the transfer of hydrogen atoms [31].

#### 3.2.1. The 2,2-Diphenyl-1-picrylhydrazyl Radical Scavenging Method

Table 1 illustrates the results of the anti-oxidant activities by means of DPPH free radical scavenging activity. The IC_50_ of honeybush tea (54.98 µg/mL) was higher than that of bush tea (20.82 µg/mL), showing the lowest anti-oxidant activity, while special tea and its blends with bush tea (BS50:50 and BS25:75) showed high anti-oxidant activity, exhibiting IC_50_ values of 2.74 µg/mL, 1.33 µg/mL and 0.62 µg/mL, respectively. The higher anti-oxidant activity of special tea may be due to a higher total phenolic and flavonoid content, as indicated in the preliminary study of the current research [38]. The same study reported that honeybush has the lowest total phenolic content, thus confirming the lower anti-oxidant activity exhibited by honeybush tea in this study.

The anti-oxidant activity of special tea using the trolox equivalent anti-oxidant capacity assay, was also confirmed by Mamphiswana et al. [39]. The overall potency of the tea samples and positive control, starting with the most potent, was: BS25:75 > vitamin C > BS50:50 > trolox > S100 > BR25:75 > BH75:25 > BR75:25 > BH25:75 > R100 > BS75:25 > BR50:50 > BH50:50 > B100 > H100. Honeybush tea exhibited lower anti-oxidant activity in this study. These results are consistent with the findings of Joubert [22] who reported that honeybush tea had lower anti-oxidant activity than rooibos and Camellia sinensis.

#### 3.2.2. The 2,2-Azino-Bis (3-Ethylbenzothiazoline-6-sulfonic Acid) Diammonium Salt Decolourisation Assay

The most effective radical scavengers (Table 1) were BS25:75 (IC_50_ = 0.26 µg/mL) and BS50:50 (IC_50_ = 0.39 µg/mL), followed by BR75:25 (IC_50_ = 0.99 µg/mL) and special tea (IC_50_ = 1.05 µg/mL). The above tea and bush tea blends’ values revealed higher anti-oxidant activity than vitamin C and trolox. Honeybush tea (IC_50_ = 23.03 µg/mL), BH50:50 (IC_50_ = 6.25 µg/mL) and BH25:75 (IC_50_ = 7.44 µg/mL) were less potent than vitamin C (1.17 µg/mL) and trolox (1.16 µg/mL). The lower anti-oxidant activity of honeybush tea could be attributed to the loss incurred during the fermentation process, as only fermented tea was used in this study [22]. Fermentation of tea leaves decreases total phenols and anti-oxidant activities, such that unfermented teas will have higher anti-oxidant activities compared to fermented teas [40]. The result is comparable with the DPPH free radical assay as honeybush tea was found to have the lowest anti-oxidant activity while the highest anti-oxidant activity was observed in special tea and its blend with bush tea.

In contrast to rooibos tea alone, the BR75:25 blend was found to have high potency as a free radical scavenger. During blending, the interaction between total phenols and tea components could result in either synergistic or antagonistic effects [40]. The effect is determined by the tea extract’s weight ratio, pH, and type of solvents used [40]. It is thus essential to determine the weight ratio that will result in a mixture with good anti-oxidant activities. The best bush tea blends identified in the current study was BS50:50, followed by BS25:75. All teas were reported elsewhere to have compounds with anti-oxidant properties such as orientin and justicidin B [41,42]. More research is needed to test the anti-oxidant properties of isolated compounds from bush and special tea, as the crude extract of these teas showed more potent anti-oxidant activity. These blends warrant further research as they hold the potential for commercialisation.

### 3.3. Anti-Inflammatory (15-Lipoxygenase) Activity

The results obtained indicate that it was only the aqueous extract of special tea (S100) (IC_50_ = 6.54), BS50:50 (IC_50_ = 8.22 µg/mL) and BS25:75 (IC_50_ = 8.98 µg/mL) which showed higher 15- lipoxygenase inhibitory activity when compared to quercetin (IC_50_ = 24.65 µg/mL) (Table 2). In a previous study conducted by Padayachee [26], essential oils from bush tea revealed anti-inflammatory activities against the 5-lipoxygenase enzyme (IC_50_ = 25.68 µg/mL). Furthermore, Padayachee [26] reported no anti-inflammatory activity on the methanol extract of bush tea at a concentration of 100 µg/mL. Although bush tea showed no anti-inflammatory effect, its blend with special tea exhibited an anti-inflammatory effect greater than the positive control. The tea extracts of the remaining tea samples were inactive against the enzyme (IC_50_ > 100 µg/mL), suggesting that the aqueous extracts could not display anti-inflammatory activity along this pathway; this warrants further studies involving other anti-inflammatory pathways such as inhibition of enzymes (cyclooxygenase-1, cyclooxygenase-2,5-lipoxygenase) and tumour necrosis factor-alpha-induced activation of nuclear factor-kappa B (NF-kB). Most plants with anti-oxidant activity also possess anti-inflammatory capacity [43]. In this study, special tea and bush tea-special tea blends exhibited the highest anti-oxidant activity, similar to the observation regarding anti-inflammatory properties. Anecdotal evidence on special tea shows that it is used as a cure for inflammation [44]. This study confirmed this result as special tea was found to have a higher anti-inflammatory capacity. Furthermore, Malongane et al. [42] reported the presence of orientin and vitexin in special tea. These two compounds were reported to have anti-inflammatory effects [45]. More research is warranted to confirm the anti-inflammatory properties of special tea using other anti-inflammatory pathways.

### 3.4. Anti-Diabetic Assay

Figure 2 depicts the in vitro assay results of the 13 selected tea samples using C2C12 myotubes. The glucose utilisation activity of the C2C12 cells are expressed as percentage of untreated (control) cells. The glucose utilisation of C2C12 muscle by tea samples (bush tea, special tea, honeybush tea and rooibos tea) at the high concentration of 500 µg/mL are not significantly different (*p* = 0.7). While special tea (*p* = 0.08) did not enhance a significant concentration dependent glucose utilisation of C2C12, bush tea (*p* = 0.003) significantly enhanced superior concentration dependent glucose utilisation. This was followed by honeybush tea (*p* = 0.03) and rooibos tea (*p* = 0.04). However, a comparison of the glucose utilisation of C2C12 muscle cells by the tea samples and blends showed further significant difference (F_0.05,5,150_ = 54.71, *p* = 0.0008) in glucose utilisation at all concentrations. The results also highlighted that samples R100 and B100 significantly (*p* ≤ 0.05) increased glucose utilisation of the myotubes at a concentration of 125 μg/mL by 26.55 ± 1.65% and 25.44 ± 2.26%, respectively, compared to BH50:50 and S100, which enhanced the lowest glucose utilisation percentage of 8.69 ± 8.69% and 10.14 ± 1.34%, respectively, at the same concentration. The positive control (insulin) at the concentration of 0.1 U enhanced C2C12 glucose utilisation activity by 43.25% (Figure 2). A similar study was conducted by Chellan [20] who reported that the aqueous crude extract of bush tea increased glucose uptake in C2C12 cells by 183.4%, 228.3% and 161.7% at concentrations of 0.03 µg/µL, 0.05 µg/µL and 0.1 µg/µL, respectively. The differences observed in percentage glucose utilisation potential of the cells in the present study and the study by Chellan [20] may be due to the high concentration of up to 100,000 µg/mL they used. Another reason for the differences may be partly due to the incubation medium. This study used RPMI, while the Chellan [20] study used Gibco Dulbecco’s Modified Eagle Medium (DMEM). The results of all tea extracts showed lower hypoglycaemic effects compared to insulin in all of the concentrations tested. The consumption of the herbal teas under this study offers a positive contribution to glucose utilisation by myotubes, although to a lesser extent than the positive control (insulin). Among the compounds previously annotated from the selected teas were mangiferin and hesperidin, and these compounds have been reported to have anti-diabetic properties [42,46].

## 4. Conclusions

The study provides scientific evidence supporting the use of rooibos, honeybush tea, special, and bush tea as potential anti-inflammatory, anti-oxidant and anti-diabetic agents, respectively. This study highlighted the potential benefit of the two indigenous teas (bush tea and special tea). For instance, the study revealed the anti-inflammatory effect of special tea, which was much higher than all the other herbal teas, supporting the anecdotal evidence that special tea is used as an anti-inflammatory agent. Furthermore, bush tea showed superior concentration-dependent glucose utilisation. The potentiation effect of blending herbal teas was highlighted, for example, a blend of special tea and bush tea showed a high anti-inflammatory effect compared to bush tea alone. Moreover, blending bush tea with the other three selected herbal teas increased its anti-oxidant activities, thus suggesting that blending of bush tea and special provides the insight for further research of its nutraceuticals potential. Further research is necessary to isolate, identify and characterize the bioactive compounds responsible for the biological activities of the selected herbal teas, particularly the special tea as it exhibited good anti-inflammatory activities.

## Figures and Tables

**Figure 1 foods-11-02233-f001:**
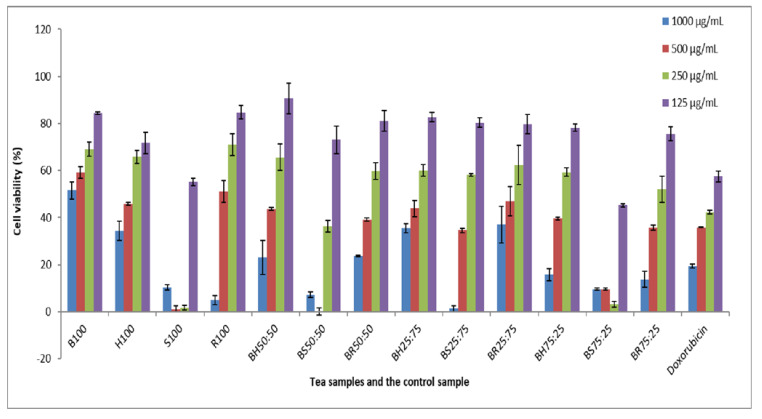
Results of cell viability of Vero cells (African green monkey kidney cell line) after treatment with selected tea samples. *Y*-axis shows percentage cell viability at different concentration. *X*-axis represents different tea samples together with the control.

**Figure 2 foods-11-02233-f002:**
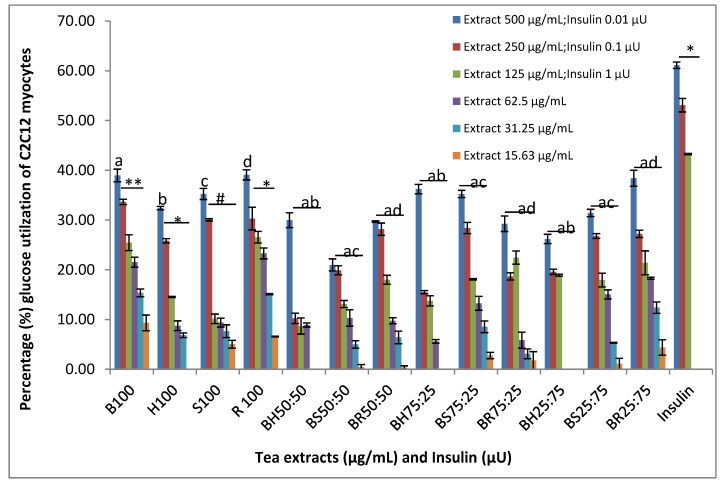
Glucose utilization activity of C2C12 myocytes (expressed as a percentage of untreated control cells ± standard error of the mean, *n* = 9) exposed to tea extracts and insulin as a positive control (1 µU = 0.05 µg/mL). Key: double asterisks indicate statistically significant difference, single asterisks and letters a, b, c, d, indicate significant difference (*p* ˂ 0.05) while (#) indicates no significant difference among tea extracts and blend. Whereas a bar beneath symbols (**,*) indicate statistical significance (*p* ˂ 0.05) within experimental groups at each point, the one beneath symbol (#) indicates no statistical significance within the experimental group at each point while those beneath letters (a, b, c, d) indicate statistical difference (*p* ˂ 0.05) between the data marked by these letters.

**Table 1 foods-11-02233-t001:** Anti-oxidant activity of selected herbal teas assessed using DPPH and ABTS decolourisation methods.

Tea Name and Positive Control (1 µg/mL)	Tea Code	DPPH ± SD (IC_50_)	ABTS ± SD (IC_50_)
Bush tea	B100	20.82 ± 3.06	4.46 ± 2.79
Honeybush tea	H100	54.98 ± 2.11	23.03 ± 2.70
Special tea	S100	2.74 ± 1.91	1.05 ± 1.34
Rooibos tea	R100	6.49 ± 2.40	2.77 ± 2.74
Bush tea 50%:Honeybush tea 50%	BH50:50	10.97 ± 1.12	6.25 ± 4.19
Bush tea 50%:Special tea 50%	BS50:50	1.33 ± 0.21	0.39 ± 0.54
Bush tea 50%:Rooibos tea 50%	BR50:50	7.74 ± 3.54	4.22 ± 3.13
Bush tea 25%:Honeybush tea 75%	BH25:75	6.28 ± 3.52	7.44 ± 2.19
Bush tea 25%:Special tea 75%	BS25:75	0.62 ± 0.68	0.26 ± 0.36
Bush tea 25%:Rooibos tea 75%	BR25:75	3.74 ± 0.592	2.11 ± 1.90
Bush tea 75%:Honeybush tea 25%	BH75:25	4.62 ± 0.38	3.03 ± 2.24
Bush tea 75%:Special tea 25%	BS75:25	6.62 ± 1.25	2.88 ± 1.83
Bush tea 75%:Rooibos tea 25%	BR75:25	4.66 ± 1.02	0.99 ± 1.03
Vitamin C		1.05 ± 0.59	1.17 ± 0.62
Trolox		1.52 ± 0.98	1.16 ± 0.25

DPPH: 2,2-diphenyl-1-picrylhydrazyl. ABTS: 2,2-azino-bis (3-ethylbenzothiazoline-6-sulfonic acid) diammonium salt.

**Table 2 foods-11-02233-t002:** Anti-inflammatory activity of selected herbal teas.

Tea Name and Positive Control (1 mg/mL)	Tea Code	15-LOX ± SD (IC_50_)
Bush tea	B100	>100
Honeybush tea	H100	>100
Special tea	S100	6.54 ± 0.84
Rooibos tea	R100	>100
Bush tea 50%:Honeybush tea 50%	BH50:50	>100
Bush tea 50%:Special tea 50%	BS50:50	8.22 ± 3.21
Bush tea 50%:Rooibos tea 50%	BR50:50	>100
Bush tea 25%:Honeybush tea 75%	BH25:75	>100
Bush tea 25%:Special tea 75%	BS25:75	8.98 ± 1.19
Bush tea 25%:Rooibos tea 75%	BR25:75	>100
Bush tea 75%:Honeybush tea 25%	BH75:25	>100
Bush tea 75%:Special tea 25%	BS75:25	64.77 ± 1.50
Bush tea 75%:Rooibos tea 25%	BR75:25	>100
Quercetin		24.65

15-LOX: 15-lipoxygenase.

## Data Availability

The data in this study are available in the article.

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
