# Peer review of "Anti-Inflammatory, Anti-Diabetic, Anti-Oxidant and Cytotoxicity Assays of South African Herbal Teas and Bush Tea Blends"

_foods, 2022, doi:10.3390/foods11152233_

Round 1

Reviewer 1 Report

General comments: The manuscript is very interesting and fits in the scope of the Foods journal. The objective of this study was to determine the antioxidant, anti-inflammatory, and antidiabetic effects and cytotoxicity of the aqueous extracts of blending bush with honeybush, rooibos and special tea. In addition, the researchers analyzed the potential synergistic effect of bush tea, honeybush, specialty tea and rooibos tea in different proportions. The manuscript is very carefully written. I read the manuscript with real pleasure. However, some changes are required.

Specific comments:

  1. Abstract: written appropriately
  2. Key words: should be added „tea”
  3. Introduction: presents the topic in an appropriate way. The authors made a good case for the point of the study.
  4. Materials and methods: described in great detail
  5. Results and discussion: Well described and discussed results. Chapter rightly divided into subchapters:  Cytotoxicity assays, Anti-oxidant activities, Anti-inflammatory activity and Anti-diabetic assay. The Authors thoroughly discussed the subject and fully achieved the assumed aim of the study. However, significant statistical differences are described in the text, but not reported anywhere (tables, figures)
  6. Figures and tables: please mark statistically significant differences;

Fig. 1. and Fig. 2.- define abbreviations under the tables

  1. Conclusions: The conclusions are a repetition of the results. They should be restructured. The authors should write what the actual conclusions of the study are: (1) whether the teas studied can be effective and what evidence there is for that, and (2) which of the teas studied and their blends is the most effective and what evidence there is for that.

8. References: I believe that relevant literature was used in the study to explore the issues fully. Recent, approx. 80% of publications come from the last 10 years. References should be formatted according to editorial requirements.

Author Response

Response to Reviewer 1 Comments

Point 1: General comments: The manuscript is very interesting and fits in the scope of the Foods journal. The objective of this study was to determine the anti-oxidant, anti-inflammatory, and antidiabetic effects and cytotoxicity of the aqueous extracts of blending bush with honeybush, rooibos and special tea. In addition, the researchers analyzed the potential synergistic effect of bush tea, honeybush, specialty tea and rooibos tea in different proportions. The manuscript is very carefully written. I read the manuscript with real pleasure. However, some changes are required.

Response 1: We appreciate the positive comment made by the reviewer.

Point 2:  Abstract: written appropriately

Response 2: Thank you for this positive feedback  

Point 3:  Introduction: presents the topic in an appropriate way. The authors made a good case for the point of the study.

Response 3: Again, thank you for the positive feedback  

Point 4:  Materials and methods: described in great detail

Response 4: We appreciate your appreciation in this section

Point 5 :  Results and discussion: Well described and discussed results. Chapter rightly divided into subchapters:  Cytotoxicity assays, Anti-oxidant activities, Anti-inflammatory activity and Anti-diabetic assay. The Authors thoroughly discussed the subject and fully achieved the assumed aim of the study. However, significant statistical differences are described in the text, but not reported anywhere (tables, figures)

Response 5: We have looked into the manuscripts and corrected the figure as suggested. We have included details of significant statistical differences and provided an explanation on the figure caption in lines 364 – 373.     

Point 6:  Figures and tables: please mark statistically significant differences; Fig. 1. and Fig. 2.- define abbreviations under the tables.

Response 6: We have managed to mark the statistically significant difference in Fig.1. We have inserted the meaning of the abbreviations under the two tables; see lines 336, 309 and 310.  

Point 7:  Conclusions: The conclusions are a repetition of the results. They should be restructured. The authors should write what the actual conclusions of the study are: (1) whether the teas studied can be effective and what evidence there is for that, and (2) which of the teas studied and their blends is the most effective and what evidence there is for that.

Response 7: The authors have rephrased the conclusion as suggested by the reviewers.  

Point 8:  References: I believe that relevant literature was used in the study to explore the issues fully. Recent, approx. 80% of publications come from the last ten years. References should be formatted according to editorial requirements.

Response 8: We have noted the comment and have checked the references to conform to the editorial requirements.  

Reviewer 2 Report

This paper deals with the investigation of anti-inflammatory, anti-diabetic, anti-oxidant and cytotoxicity assays of south African herbal teas and bush tea blends. The paper is written correctly, the experiments were set properly and the conclusions arise from the results. However, the paper lacks novelty and key explanations.

Major shortcomings:

How do you explain the increased antioxidant activity of tea blends?

What are the mechanisms behind investigated biological activities?

Which compounds are responsible for the investigated biological activities?

Minor shortcoming:

Line 113 Which freeze drier was used?

Author Response

Response to Reviewer 2 Comments

Point 1:  This paper deals with the investigation of anti-inflammatory, anti-diabetic, anti-oxidant and cytotoxicity assays of South African herbal teas and bush tea blends. The paper is written correctly, the experiments were set properly and the conclusions arise from the results. However, the paper lacks novelty and key explanations.

Response 1: The manuscript has sufficient novelty as the biological activities of special tea were not previously investigated before this study. In addition, blending bush tea and special tea to investigate potential useful biological activities was not done. The positive results that are shown by blending bush tea and special tea will stimulate further research on these indigenous teas.  

Point 2:  How do you explain the increased anti-oxidant activity of tea blends?

Response 2: The increased anti-oxidant activity of the blends can be attributed to the synergistic activity of the teas. For instance, special tea had a good anti-oxidant activity alone, but the anti-oxidant effect was higher in combination with bush tea. The underlying mechanism of action was not investigated, and this will be looked into in further research.  

Point 3:  What are the mechanisms behind investigated biological activities?

Response 3: We did not investigate the mechanism, and this aspect will be looked into in further research.

Point 4:  Which compounds are responsible for the investigated biological activities?

Response 4: We have published the chemical analysis of the four teas; for more information, please refer to the article "Malongane, F., McGaw, L. J., Nyoni, H., & Mudau, F.N. (2018). Metabolic profiling of four South African herbal teas using high-resolution liquid chromatography-mass spectrometry and nuclear magnetic resonance. Food Chemistry, 257 (September 2017), 90–100. https://doi.org/10.1016/j.foodchem.2018.02.121". In this study we have, however, determined the action of the crude extract on the biological activities and not individual compounds. 

Point 5:  Line 113 Which freeze drier was used?

Response 5: The name of the freeze drier is inserted in line 113 as Labconco Freezone 4.5 Benchtop Freeze Dryer

Reviewer 3 Report

The authors analysed the anti-oxidant, anti-inflammatory and anti-diabetic properties of several teas consumed in South Africa.

The article is interesting and well-organized, and the results are clearly presented. There are, however, minor issues that need to be solved.

Line 38: Please change loss of voice to aphonia.

Line 83 to 91: Although the objectives of the work are presented, it is not clear why the authors prepared a mixture of bush tea with all the other teas and did not prepare combinations of special tea with rooibos tea, for instance. Please explain this choice.

Cytotoxicity assays: The results show that Special tea is cytotoxic at higher concentrations but less toxic at 1000 than at 500 and 250 μg/mL. The same effect was observed when mixing this tea with bush tea. Are these results correct? Is there an explanation for this observation?

Figure 1 caption. Please correct the letter size of the caption. 

Line 309: Please indicate the author's name before the citation number.

Figure 2: The legend of the Y-axis is in red. Please correct it to black.

Author Response

Response to Reviewer 3 Comments

Point 1:  The authors analysed the anti-oxidant, anti-inflammatory and anti-diabetic properties of several teas consumed in South Africa.

The article is interesting and well-organized, and the results are clearly presented. 

Response 1: We thank the reviewer for the positive response

Point 2:  Line 38: Please change loss of voice to aphonia.

Response 2: As suggested, the author changed the loss of voice to aphonia.  

Point 3:  Line 83 to 91: Although the objectives of the work are presented, it is not clear why the authors prepared a mixture of bush tea with all the other teas and did not prepare combinations of special tea with rooibos tea, for instance. Please explain this choice

Response 3: This study is part of a departmental study promoting the use of bush tea. The study was undertaken to expand on the potential benefits of bush tea, especially when blended with other teas. Thus bush tea formed the basis of the study.  

Point 4:  Cytotoxicity assays: The results show that Special tea is cytotoxic at higher concentrations but less toxic at 1000 than at 500 and 250 μg/mL. The same effect was observed when mixing this tea with bush tea. Are these results correct? Is there an explanation for this observation?

Response 4: Although bush tea is less cytotoxic compared to special tea, the mixing of these teas did not lessen the effect of special tea as being cytotoxic. We have checked the raw data from each of the replicated experiments and confirm that in the case of special tea, alone and when combined with bush tea, the highest concentration of 1000 μg/mL led to slightly higher cell viability than at 500 and 250 μg/mL, but we believe that this is possibly owing to cell growth patterns or unexpected effects of the extract on the cells which would have to be investigated in more depth, perhaps microscopically. At this stage, we cannot speculate on a reason for this occurrence.

Point 5:  Figure 1 caption. Please correct the letter size of the caption. 

Response 5: We thank the reviewer for this correction; the letter size was corrected as suggested.  

Point 6:  Line 309: Please indicate the author's name before the citation number.

Response 6: The name of the author was inserted as suggested.  

Point 7:  Figure 2: The legend of the Y-axis is in red. Please correct it to black.

Response 2: We thank the reviewer for this observation; we have changed the colour of the Y-axis.  

Round 2

Reviewer 2 Report

can be accepted in the present form